# The Effectiveness of a Leadership Subject Using a Hybrid Teaching Mode during the Pandemic: Objective Outcome and Subjective Outcome Evaluation

**DOI:** 10.3390/ijerph19169809

**Published:** 2022-08-09

**Authors:** Wenyu Chai, Xiang Li, Daniel T. L. Shek

**Affiliations:** Department of Applied Social Sciences, The Hong Kong Polytechnic University, Hong Kong, China

**Keywords:** positive youth development, university students, leadership program, higher education, hybrid mode teaching, COVID-19 pandemic

## Abstract

Positive youth development (PYD) is an innovative approach to protect students from mental health problems and promote their positive and holistic development. Although there are many studies on the beneficial effects of PYD programs on youth in high school and community contexts, it is not clear whether subjects adopting PYD principles can promote positive development for university students. Moreover, it is unclear whether such subjects are effective under COVID-19, where subjects are commonly taught via the “hybrid” mode (i.e., face-to-face plus online teaching). The present study examined students’ changes in the PYD, wellbeing, and desired graduate attributes after they had taken a leadership subject utilizing PYD principles taught by the “hybrid” mode (*N* = 630). Adopting the one-group pre-test and post-test design (i.e., objective outcome evaluation), we found that students showed significant positive improvement in PYD indicators, wellbeing, as well as desired graduate attributes. Additionally, students had high satisfaction with the course design and teaching staff, and perceived many benefits from this subject based on the subjective outcome evaluation conducted at the end of the subject. Results also showed that students’ satisfaction with the curriculum significantly and positively predicted their positive change in PYD indicators, indicating the convergence of subjective outcome evaluation and objective outcome evaluation. The results highlight the positive impacts of the hybrid mode leadership subject with PYD principles in higher education.

## 1. Introduction

### 1.1. Challenges Faced by the Higher Education Sector

In recent years, there has been increasing concern for the positive and holistic development of university students due to both internal and external challenges faced by higher education [1,2,3]. Regarding internal challenges, the transition to university is a significant challenge as students need to deal with multifaceted adjustment issues, such as new academic and social environments, high independency, different learning modes, being away from home, peer pressure, and academic and financial demands [4,5,6,7]. These adjustment difficulties would also expose students to a higher risk of mental health problems [4]. In fact, research consistently showed high prevalence rates of mental health problems (e.g., depression, anxiety, and psychological distress) among university students, which were substantially higher than the general population [8,9,10,11]. Poor mental health would lead to other negative consequences, such as an increase in problem behaviour, as well as a decrease in physical health conditions, academic performance, and life satisfaction [12,13,14]. Therefore, it is important to develop effective preventive programs for university students. One possible direction is to promote the developmental assets of young people because research indicated that psychological strengths and positive qualities could promote students’ well-being (e.g., self-esteem, self-efficacy, resilience, optimism, social emotional competence), and buffer the negative impact of different challenges and pressures on student development [4,15,16].

In addition to internal challenges, higher education is also facing challenges posed by the changing economic and societal conditions. One major social change is the transformation from a manufacturing economy to a service economy in recent decades [17]. Although the manufacturing economy focuses on standardized production and requires semi-skilled or skilled manual workers, the service economy depends on knowledge, information, innovation, and requires holistic qualities of employees [18,19]. To produce high-quality services, intrapersonal and interpersonal competences, such as effective communication, problem solving, critical thinking, and creativity, are significantly required [17,20]. In addition, knowledge explosion and rapid advancement of science and technology make work and life “uncertain” and “ever-changing” [18], which also requires high adaptability, flexibility, lifelong learning, as well as capacities to deal with challenges such as resilience and emotional competence [18,21]. Hence, it is important for higher education to find effective means to develop these intrapersonal and interpersonal qualities and to promote holistic development of university students.

### 1.2. Positive Youth Development (PYD) Approach

To find effective means to prevent students from developing mental health problems and promote their holistic development, the positive youth development (PYD) framework is a promising approach. As an innovative paradigm in psychology and youth studies, PYD focuses on the development of strengths and potentials of youths, which is different from the pathological orientation of traditional psychological research and youth studies [22,23,24]. Different theories and models of PYD were proposed by scholars. Lerner and his colleagues [25] conceptualized PYD as a developmental process achieved through positive interaction between individual and his/her environments. If the environment is constructed to promote the development of strengths of adolescents, PYD occurs which could be indicated by 5 qualities, namely “competence”, “confidence”, “connection”, “character”, and “caring” [25]. Catalano and his colleagues [23] proposed 15 indicators of PYD after studying different PYD programs in the United States. These 15 indicators include “cognitive competence”, “social competence”, “emotional competence”, “clear and positive identity”, “resilience”, “spirituality”, and prosocial norms. Benson and his colleagues [22] proposed the concept of developmental assets under the framework of PYD which includes internal and external assets. The internal assets refer to internal positive qualities or competences including “commitment to learning”, “positive values”, “social competencies”, and “positive identity”; while the external assets refer to quality relationships received from the environment including “support”, “empowerment”, “boundaries”, and “expectations” from parents, peers, and community, and “constructive use of time” [26] (p. 29). Although there are differences among these perspectives, they all share a common idea that intervention programs should be developed to promote the development of positive qualities and strengths to protect students from problems and foster their positive development and thriving [27].

In line with the theoretical postulations, empirical findings showed that higher levels of PYD predicted lower levels of mental health problems indexed by depression, anxiety, emotional problems, hopelessness, and suicidal ideation [28,29,30]. Higher levels of PYD also predicted lower levels of problem behaviour, such as delinquency and internet addiction [31,32], and higher levels of healthy functioning, academic performance, and academic wellbeing [33,34]. Taken as a whole, both theoretical models and empirical findings suggest that PYD is a potentially effective approach to foster holistic development of university students to help them meet both internal and external challenges.

### 1.3. PYD and Wellbeing of University Students

Among different correlates and outcomes of PYD, research consistently showed that PYD attributes were positively linked to subjective wellbeing, such as life satisfaction [35] and thriving [26]. For example, research on Chinese adolescents showed that PYD positively predicted life satisfaction among secondary school students in Hong Kong in both cross-sectional and longitudinal studies [36,37]. In addition, PYD indicators, such as the five “Cs” and developmental assets, also significantly and positively predicted youth thriving [26,38]. Thriving refers to an ideal status, manifestation, exemplar, outcome, or process of positive youth development [22,39,40]. It is manifested by adolescents’ active participation, commitment, and contribution to their community and society [39]. Scholars argued that adolescents’ development of the PYD indicators, such as the five “Cs” or developmental assets, would finally contribute to thriving [21,40].

### 1.4. PYD Programs and Related Research

Along with its theoretical development and empirical studies, PYD models have been adopted and incorporated in many programs in educational and social service sectors. There is also a body of literature showing positive effects of these programs [41,42]. For example, a systematic review of 23 school-based PYD programs showed that these programs had significant and positive impact on students’ development of different psychological strengths and wellbeing [41]. Particularly, the review showed that curricula- and lecture-based PYD programs promoted students’ social and emotional skills and other aspects of positive development. In addition, a meta-analysis on the effects of 82 school-based social emotional learning interventions based on the PYD approach revealed that these interventions significantly promoted students’ social and emotional competence, as well as other indicators of wellbeing [43]. Furthermore, a meta-analysis on 24 outside-school PYD interventions showed that, regardless of program features, these programs had a significant and positive impact on adolescents’ psychological adjustment and academic performance although with a small effect size [44]. However, these programs had no effect on promoting positive social behaviours and reducing problem behaviours. Moreover, several studies showed that PYD programs promoted the PYD attributes and life satisfaction, as well as reduced depression and problem behaviours in Chinese secondary school students [32,45].

### 1.5. Research Gaps

There are several research gaps in the existing literature in this field. First, most of the existing studies mainly focused on PYD programs and their effects in high school and community contexts, with very few studies on university students, who are an important group of youths or emergent adults. Second, there is inadequate research on the application of PYD programs in leadership programs (such as credit-bearing subjects) in a non-Western context. Based on a review of research on PYD from 1995 to 2020 [46], one major research gap identified was the dominating position of Western studies in PYD research. In another review of PYD studies in the Asian context, it was found that there were very few validated PYD programs in Asian communities with rigorous evaluation design [47]. Therefore, there is an urgent need for research on PYD programs and their effects in the Asian context.

Third, there are a few studies on the effects of educational programs based on the PYD framework under the COVID-19 which adopt the hybrid and online modes of teaching [48,49,50,51]. The pandemic has changed the traditional format of teaching and learning in many higher education institutions. To cope with challenges of the pandemic, many institutions have to adjust their teaching and learning approach from the traditional face-to-face mode to either pure online mode or a hybrid mode which allows students to continue their study [52,53]. Particularly, hybrid-mode teaching has become a widely adopted practice in higher education during the COVID-19 pandemic. Although there are different definitions of hybrid-mode teaching, it, in a broader sense, refers to an integration of online learning (synchronized/asynchronized) with face-to-face teaching [54]. In a narrow sense, scholars argued that hybrid-mode teaching focuses more on conducting online synchronized teaching and face-to-face teaching simultaneously in the same class session to accommodate the needs of students in different locations [55], which has been named as “HyFlex” teaching.

However, there are conflicting perceptions on the effects of hybrid-mode teaching. Some scholars argued that the hybrid-mode teaching had a lot of benefits, such as allowing high flexibility, more student choices, and catering different needs [55]. However, some scholars argued that the hybrid-mode teaching could have negative effects on student learning, such as enlarged differences in students’ learning experience and undermining class and group cohesion and students’ sense of belonging [54]. Unfortunately, there are limited empirical studies on the effect and impact of hybrid-mode teaching [51]. Some recent studies showed that students encountered challenges in online learning, such as their lack of readiness, as well as physical and psychological preparation for the online learning [56]. Nevertheless, there are also studies showing that students performed better during the pandemic due to the increased self-autonomy in learning strategies [57] and they preferred the flexibility in online and hybrid learning [52]. Under the pandemic, it is urgent to understand the effects of PYD program adopting hybrid-mode on students’ development.

Fourth, there is a need to understand the impacts of PYD programs based on rigorously designed evaluation studies [47]. Particularly, there is a need to include both objective outcome evaluation and subjective outcome evaluation to provide a more comprehensive understanding of the program effects and there is a need to advance our understanding of the association between objective outcome evaluation and subjective outcome evaluation. Subjective outcome evaluation is an important and commonly used evaluation method in educational, health care, and social service fields [58,59]. It provides important information on the perceptions, feelings, and viewpoints of program participants who are important stakeholders and beneficiaries of intervention programs. However, this evaluation method was criticized by some scholars as being biased and not related to objective evaluation outcomes [60]. Additionally, there are inconsistent empirical findings. Some studies found no or a weak correlation between subjective measures and objective measures [61,62], while some studies showed that subjective outcome evaluation results were significantly correlated with objective outcome evaluation results [63,64,65]. Therefore, there is a strong need to further test the relationship between subjective outcome evaluation and objective outcome evaluation findings to advance understanding in this area to determine the effectiveness of educational and intervention programs.

### 1.6. A Leadership Program Based on PYD Framework in a Local University in Hong Kong

Promotion of holistic development among university students is an urgent task for higher education institutions in Hong Kong. Similar to the international landscape, there is a high prevalence of mental health problems among Hong Kong university students. For example, in a recent survey on 1200 undergraduate students from the eight public universities in Hong Kong, 68.5% students reported mild to severe depression symptoms associated with mild to severe anxiety symptoms [66]. In another recent study, the prevalence rate of depression among university students in Hong Kong was much higher (41.0%) than those in mainland China (16.8%) and Macau (35.2%) [67]. In addition, the dominance of the service economy in Hong Kong raises requirements for university graduates with strong intrapersonal and interpersonal competences [68,69]. Against this background, a leadership program named “Tomorrow’s Leaders” (TL) was developed based on the PYD framework in one public university in Hong Kong to facilitate students’ positive and holistic development. The TL program is a three-credit compulsory subject for all the first-year undergraduates enrolled in the university. It aims to equip students with important concepts and theories of PYD qualities (i.e., intrapersonal and interpersonal qualities) that are essential for being future leaders of the society. Specifically, the subject has six intended learning outcomes: (1) “understand and integrate theories, research and concepts on the basic qualities (particularly intrapersonal and interpersonal qualities) of effective leaders”; (2) “develop self-awareness and self-understanding”; (3) “demonstrate self-leadership in pursuit of continual self-improvement”; (4) “apply intrapersonal and interpersonal skills in daily lives”; (5) “appreciate the importance of intrapersonal and interpersonal qualities in effective leadership”; and (6) “recognize and accept their responsibility as professionals and citizens to the society and the world”. These intended learning outcomes are also aligned with the desired graduate attributes of the university, such as life-long learning, problem solving, critical thinking, and ethical leadership. With reference to the objectives, the content of the subject focuses on 9 essential intrapersonal and interpersonal qualities delivered in 13 lectures including “self-leadership”, “social emotional competence”, “resilience and stress-coping”, “morality”, “spirituality”, “law-abiding leadership”, “cultural competence and global citizenship”, “effective communication”, and “team building”. For each lecture, there are group discussion, reflective exercises, role play, case discussion to facilitate students’ experiential learning, active learning, and self-reflection to link and apply the learned quality in daily life and future profession. In addition, three assessment strategies (i.e., class participation, group project, and term paper) are used to help students consolidate their learning experience.

Since its implementation from 2012 to 2019, the subject was delivered in face-to-face mode. Different evaluation studies showed that the subject was effective in promoting students’ development in PYD qualities and wellbeing [70,71,72]. However, due to the COVID-19 pandemic outbreak in Hong Kong, most of the universities adopted either pure online teaching or hybrid teaching mode. In accordance with the university policy, the TL subject transformed its teaching and learning from face-to-face mode to a hybrid-mode in Semester 1 of the 2021/22 academic year. Specifically, for each lecture, the original three-hour face-to-face lecture was changed to a two-hour lecture plus one-hour non-synchronized before-class online self-regulated learning. For the two-hour lecture, students could attend the lecture either in face-to-face approach or online approach (i.e., HyFlex mode) based on their preferences and availability.

### 1.7. The Present Study

Although the traditional face-to-face mode TL subject was shown to be effective in promoting students’ development in PYD qualities and wellbeing, whether the subject is still effective by adopting the hybrid-mode teaching under the COVID-19 is unknown. Hence, the present study aimed to examine the effects of the hybrid-mode TL subject. With reference to the above literature review, three research questions and related hypotheses are proposed:Do students change after taking the TL subject with PYD focus via the hybrid mode of teaching and learning? Based on the past studies [70,71], we expected that students would show positive changes in the outcome indicators, including PYD attributes (Hypothesis 1a), well-being (Hypothesis 1b) and desired graduate attributes (Hypothesis 1c);Are students satisfied with their learning experience, including perceptions of course design, teaching staff, and benefits of the subject delivered via the hybrid mode? With reference to past studies [73,74], we hypothesized that a majority of the students would have positive perceptions of the course design, teacher, and benefits of the subject (Hypothesis 2);Is students’ satisfaction with their learning experience associated with the objective outcome evaluation measures? With reference to the literature [63,75,76], we expected that students’ satisfaction would be significantly related to their outcomes over time (Hypothesis 3) in objective outcome evaluation.

## 2. Materials and Methods

### 2.1. Participants and Procedure

The one-group pre-test and post-test design was adopted in the present study. This method is widely used in different research fields, such as clinical health, education, social work, and program evaluation [77,78,79]. Many studies have adopted this method to evaluate the effects of intervention or education programs including those studies published in higher-ranking journals, such as *International Journal of Environmental Research and Public Health* [77], *Research on Social Work Practice* [78], and *Assessment and Evaluation in Higher Education* [7]. For example, Gil and Kim [77] used the one-group pre-test and post-test design to evaluate the effects of an online self-help intervention on preventing depressive symptoms. Son [79] used this design to examine the effects of an undergraduate course on nursing students’ empathy and caring for the elderly. Additionally, Chan and Holosko [78] evaluated the effectiveness of a youth media practice program on self-esteem and critical thinking using a one-group pre-test and post-test design.

Scholars argued that the method is useful and effective to understand the effects of intervention programs under several situations. First, one-group pre-test and post-test design would be adopted if it is difficult or unfeasible to form a control group [80,81]. This is a major reason for most evaluation and intervention studies that adopted this research design. For example, Kurki and his colleagues [82] adopted a one-group pre-test and post-test design to examine the effects of a digital mental health literacy program on the wellbeing of medical students due to the reason that “only one group is available” [82] (p. 10) and it is not feasible to divide students to control group and experimental group. In another study, Siu and his colleagues [83] examined the effects of two intervention studies on stress management using a one-group pre-test and post-test design due to the limit of budget and time to recruit a control group. As TL subject is a compulsory subject offered to nearly all the first-year students at this university as a requirement for graduation, it is practically infeasible to assign students to a control group who would not take the subject. Additionally, as the number of participants is large, it is quite difficult and expensive to recruit students who did not take this subject from another university as the control group. Second, one threat to the internal validity of the method is that there might be other factors influencing the results such as maturation and history (other events). However, this threat would be reduced if the duration of study is shorter [80]. As the duration of the present study was one semester (13 weeks), the possible influence of other factors (especially natural maturation) may be reduced.

The participants were recruited from the undergraduates taking the hybrid-mode TL subject in Semester 1 of the 2021/22 academic year. In the first lecture, all the students were invited to complete an online objective outcome evaluation questionnaire (pre-test). Formal consent was obtained from the participants through an online consent form. At the end of the semester, the students were invited to complete the same questionnaire again (post-test) and a subjective outcome evaluation questionnaire. A total of 630 students completed both pre- and post-test questionnaires and the subjective outcome evaluation questionnaire. Most of the participants (83.5%) were 18 years old or above and 48.2% of them were female. In addition, 62.0% of the participants were born in Hong Kong and 32.4% of them were born in mainland China. The detailed demographic information of the participants is shown in Table 1.

### 2.2. Measures

#### 2.2.1. Chinese Positive Youth Development Scale (CPYDS)

The PYD qualities were measured using a short version “Chinese Positive Youth Development Scale” (CPYDS). Developed based on the PYD attributes identified by Catalano and his colleagues [23], CPYDS measures the positive youth development of Chinese adolescents [84]. The scale possesses good psychometrical properties based on different validation studies [84,85]. The short version CPYDS adopted in the present study contains 34 items assessing 12 PYD qualities (primary factors of the scale). The 12 qualities are “resilience” (e.g., “When I face difficulty, I will not give up easily”), “social competence” (e.g., “I know how to communicate with others”), “emotional competence” (e.g., “When I am unhappy, I can appropriately show my emotions”), “cognitive competence” (e.g., “I know how to find the causes of and solutions to a problem”), “behavioural competence” (e.g., “I can face criticisms with an open mind”), “moral competence” (e.g., “I will fulfil my promise”), “self-determination” (e.g., “I am able to make wise choices”), “self-efficacy” (e.g., “I can finish almost everything that I am determined to do”), “clear and positive identity”(e.g., “I know my strengths and weaknesses”), “beliefs in the future” (e.g., “I have confidence to graduate from university”), “prosocial norms” (e.g., “It is my pleasure to obey rules and regulations”), and “spirituality” (e.g., “My life is colourful and full of excitements”). In addition, there are three higher order factors which are “cognitive-behavioural competencies”, “general PYD qualities”, and “positive identity”. Each item was rated through a six-point scale (from “1 = Strongly Disagree” to “6 = Strongly Agree”). The short-version CPYDS obtained acceptable to good internal consistency for each primary factor and higher order factor in a recent study [7]. The Cronbach’s alpha for the primary factors and higher order factors ranged from 0.72 to 0.94 in the present study (Table 2).

#### 2.2.2. Wellbeing

We used two indicators to assess student wellbeing: life satisfaction and thriving. Life satisfaction was evaluated through the “Satisfaction with Life Scale” (SWLS) [86]. SWLS contains five items assessing the global life satisfaction of people which demonstrated good psychometric properties [86,87]. Each item is evaluated using a six-point scale (from “1 = Strongly Disagree” to “6 = Strongly Agree”). A sample item is “In most ways, my life is close to my ideal”. Thriving was assessed by a scale developed based on Lerner and his colleagues’ [40] theoretical proposition of thriving as exemplified condition of PYD. The scale consists of five items evaluated through a six-point scale (from “1 = Strongly Disagree” to “6 = Strongly Agree”). The scale obtained good internal consistency in a previous study [7]. In the present study, the Cronbach’s alpha values for the scale are 0.60 and 0.78 in pre-test and post-test questionnaires, respectively.

#### 2.2.3. Desired Graduate Attributes

Four desired graduate attributes were assessed, including perceived problem-solving ability, lifelong learning, ethical leadership, and critical thinking. Problem solving was assessed by 3 items with a sample item being “I know how to effectively solve problems in my daily life”. Lifelong learning was assessed by 2 items with a sample item being “It is important to understand the development of oneself”. Critical thinking was assessed by 3 items with a sample item being “I know how to use critical thinking skills when solving problems”. Ethical leadership was assessed by 15 items with one sample item being “I treat other people with dignity and respect”. All the items are rated on a six-point scale (from “1 = Strongly Disagree” to “6 = Strongly Agree”). The scales showed acceptable to good internal consistency in a previous study [7]. The Cronbach’s alpha values for the four scales in the present study ranged from 0.69 to 0.97.

#### 2.2.4. Subjective Outcome Evaluation Scale

The participants’ satisfaction with their learning experience in TL subject was assessed by a Subjective Outcome Evaluation Scale (SOES) developed by the curriculum team based on validated measures used in previous studies [73,88]. The scale contains 41 items assessing students’ satisfaction with three aspects of their learning experience, namely their satisfaction with the course design (10 items), teacher (10 items), and benefits of the course (21 items). There are three additional items in the scale assessing students’ overall satisfaction with the course, their willingness to recommend the course to others, and their willingness to join similar courses in future. Previous studies showed that the scale possessed good psychometric properties [73,74]. The Cronbach’s alpha values for satisfaction with the course design, satisfaction with the teacher, and satisfaction with the benefits of the course were 0.97, 0.98, and 0.99, respectively, indicating excellent internal consistency.

### 2.3. Data Analyses

First, descriptive analyses were conducted to understand the demographic information of the participants. To understand the participants’ changes in PYD attributes, wellbeing, and desired graduate attributes after taking the TL subject, a one-way repeated measures multivariate analysis of variance (MANOVA) was conducted. To understand students’ satisfaction with course design, teacher, and benefits of the course, descriptive analyses were computed to calculate frequencies and percentages of students’ ratings in subjective outcome evaluation. Multiple regression was also computed to examine whether students’ satisfaction with course design, teacher, and benefits of the course predicted their overall satisfaction with the subject. To examine the association between objective outcome evaluation and subjective outcome evaluation, hierarchical regression was conducted to examine the predicting effect of students’ perceived course design, teacher, and course benefits on post-test PYD qualities after controlling pre-test PYD qualities and demographic factors. All the above analyses were conducted using SPSS Version 25.

## 3. Results

### 3.1. Objective Outcome Evaluation

One-way repeated measures multivariate analysis of variance (MANOVA) was conducted to examine students’ changes in PYD attributes, wellbeing, and desired graduate attributes from pre-test to post-test. We found significant multivariate effects of time on 12 primary factors of PYD, the four higher-order factors of PYD, wellbeing, and desired graduate attributes (*F* ranged between 18.17 and 89.99, *ps* < 0.001, η^2^_p_ ranged between 0.039 and 0.261). Results showed positive improvement in all the 12 PYD attributes, higher-order PYD qualities, life satisfaction, thriving, and four desired graduate attributes (*ps* < 0.001). Details can be found in Table 2. The findings are presented in accordance with the format commonly adopted in quantitative studies including those published in the *International Journal of Environmental Research and Public Health* [7,89].

### 3.2. Subjective Outcome Evaluation

Students had very positive perceptions of course design, teacher quality, and course benefits. The percentages of positive ratings of the participants on different items in subjective outcome evaluation questionnaire ranged from 77.8% to 88.5% for course design, 91.7% to 94.9% for teacher quality, and 73.1% to 80.8% for course benefits. Details can be found in Table 3.

Hierarchical multiple regression analyses showed that students’ perceived course design, teacher quality, and course benefits significantly predicted their perceived overall satisfaction with this subject after controlling the effects of gender, age, place of birth, and faculty (course design: *β* = 0.45, *p* < 0.001, Cohen’s *f^2^* = 0.098; teacher quality: *β* = 0.13, *p* < 0.01, Cohen’s *f^2^* = 0.020; course benefits: *β* = 0.27, *p* < 0.001; Cohen’s *f^2^* = 0.049), respectively. The findings provided support for Hypothesis 2. Details can be found in Table 4.

### 3.3. Predictive Effects of Students’ Perceptions of the Subject on Their Objective Outcomes

A two-step model based on hierarchical multiple regression was conducted to examine the relationship between subjective outcome evaluation and objective outcome evaluation. In Step 1, gender, age, place of birth, faculty, and students’ pre-test scores of PYD attributes were entered in the analyses as independent variables and students’ post-test PYD scores were entered as dependent variable. Results showed that place of birth and students’ pre-test PYD scores significantly predicted students’ post-test PYD scores (place of birth: *β* = 0.13, *p* < 0.01, Cohen’s *f^2^* = 0.019; pre-test PYD: *β* = 0.47, *p* < 0.001, Cohen’s *f^2^* = 0.271). In Step 2, students’ perceptions of course design, teacher quality, and course benefits were put in the analyses as independent variables to examine their predicting effects on students’ post-test PYD scores. Results showed that students’ perceptions of course design, teacher quality, and benefits were significantly related to their change in post-test scores in PYD attributes after the effects of gender, age, place of birth, faculty, and pre-test PYD attributes were controlled (course design: *β* = 0.21, *p* < 0.01, Cohen’s *f^2^* = 0.013; teacher quality: *β* = 0.12, *p* < 0.01, Cohen’s *f^2^* = 0.011; course benefits: *β* = 0.26, *p* < 0.001; Cohen’s *f^2^* = 0.028). The findings provided support for Hypothesis 3. Details can be found in Table 5.

## 4. Discussion

With regard to the first research question, results showed that the participants showed significant positive changes in PYD qualities, wellbeing, and desired graduate attributes from pre-test to post-test, which supports Hypotheses 1a, 1b, and 1c. The findings are in line with previous evaluation findings that students gained significant and positive changes in PYD qualities and life satisfaction after taking this subject [7,70,71,72]. Although the previous evaluation findings focused on the effectiveness of the subject in early years of the implementation, the present study indicated that the subject has sustained positive impact over time. It reinforces the thesis that this credit-bearing subject based on the PYD approach is effective in promoting the positive development and wellbeing of university students.

In addition, the findings advance our understanding of the effectiveness of this subject in promoting the well-being and desired graduate attributes of the students. This observation has implication for understanding the role of PYD programs in higher education. One significant challenge to the curricula in higher education is the difficulty in implementing effective general education program due to the over-division of knowledge and specialized disciplines [90,91]. Many general education programs were criticized as fragmented and piecemeal [92], which makes them incapable of fostering students’ holistic development. The present study suggests that the subject based on the PYD approach is a promising strategy to promote students’ holistic development and desired graduate attributes.

In addition, the findings contribute to our limited understanding of the effects of hybrid mode of teaching in higher education, particularly during the COVID-19. Although hybrid mode of teaching has become a trend or widely adopted practice in higher education under COVID-19, there is limited empirical studies on this new teaching mode [59]. There are also inconclusive findings in the existing literature. Although some studies showed benefits of this mode of teaching [52,93], other studies suggested this teaching mode may have negative influence on students’ learning motivation and outcomes [94,95]. The findings of this study contribute to this body of literature by showing the benefits and positive impact of hybrid-mode teaching on students’ learning experience and outcomes in higher education.

Regarding the second research question, results showed that the participants had a high-level of satisfaction with their learning experience in the subject conducted in the hybrid mode, including their satisfaction with course design, teacher, and benefits of the subject. The findings support Hypothesis 2, and they are consistent with previous subjective outcome evaluation findings that students were generally satisfied with the course design and teaching staff and benefited from the subject after taking the course [73,74,88]. However, as the previous evaluation studies mainly focused on the subject based on the face-to-face mode, the present findings suggest that students also showed positive perceptions of the subject conducted in hybrid mode. This is also in line with a recent study of students’ perceptions of HyFlex mode TL subject based on post-lecture evaluation, showing that students had positive views towards the subject and perceived the subject beneficial to their positive development [51]. Similar support also emerged from hybrid teaching and provision of service under the context of service-learning [50,76].

Subjective outcome evaluation is a commonly used approach in program and curriculum evaluation. Different from objective outcome evaluation focusing on students’ changes in important indicators, subjective outcome evaluation focuses on different stakeholders’ perceptions and satisfaction with different aspects of programs and their subjective experience, which offers important information for course evaluation and further improvement [32,96]. One important rationale for subjective outcome evaluation is that program participants should “have their opinions heard” as important information for program improvement [97] (p. 7). The present study showed students were highly satisfied with all the three aspects of the subject delivered via the hybrid mode.

Regarding the third research question, results showed that the participants’ satisfaction with their learning experience positively predicted their post-test PYD scores in objective outcome evaluation after the effect of demographic factors and pre-test PYD scores were controlled, hence supporting Hypothesis 3. This observation contributes to the scholarship on evaluation methodology. Argued by Shek [64], it is significant and meaningful to examine the association between objective outcome evaluation and subjective outcome evaluation because the use of subjective outcome evaluation would be meaningless if it is unrelated to intervention effects. The result sheds light on the disputes and inconclusive findings on relationship between objective and subjective outcome evaluation in existing literature. Although some studies showed a significant positive relationship between objective and subjective outcome evaluation [63,98,99], other studies showed no significant association between the two evaluation measures and outcomes [61,62]. One possible reason might be due to the subjectivity of subjective outcome evaluation [100]. Zhu and Shek [100] argued that the objectivity of subjective outcome evaluation could be increased through valid and reliable measures. Several studies based on multidimensional and validated measures of subjective outcome evaluation also showed significant association between subjective and objective outcome measures [64,65,100].

Although the present study included both objective and subjective outcome evaluation strategies to gain a more comprehensive understanding of the effects of the subject conducted via hybrid mode, several limitations should be noted. First, the study adopted one group pre-test and post-test design in objective outcome evaluation, which may have bias arising from the absence of a control group, since the participants’ improvement may be due to other factors, such as maturation, but not the participation in the subject [81]. Therefore, future research based on quasi-experimental design should be conducted to strengthen the present findings. Second, while the present study adopted objective and subjective outcome evaluation, other evaluation methods might also need to be involved, such as qualitative evaluation to triangulate the data and provide deeper understanding of underlying mechanisms for subject effectiveness. Despite of the limitations, these findings are important because COVID-19 has substantially changed the teaching and learning mode under COVID-19 [101,102]. In conjunction with recent studies reporting the value and benefits of online teaching and learning [10,48,49,103,104], the present study sheds light on the impacts and effectiveness of PYD programs conducted via a Hybrid mode in higher education as an important approach for promoting university students’ positive development, wellbeing, and holistic development.

## 5. Conclusions

There are very few studies studying the effectiveness of hybrid teaching, especially in the Chinese context under COVID-19. This study examined the effectiveness of a leadership subject utilizing positive youth development principles delivered via a hybrid teaching mode. After taking the subject, students showed positive changes and they had positive perceptions of the subject content, instructors, and benefits. We also found that students’ satisfaction with the subject predicted post-test measures of well-being and desired graduate attributes.

## Figures and Tables

**Table 1 ijerph-19-09809-t001:** Demographic information of participants (*n* = 630).

Demographic Variables	Frequency	Percent
**Gender**		
Male	326	51.8%
Female	303	48.2%
**Age**		
16 or below	1	0.2%
17	102	16.3%
18	338	54.2%
19	92	14.7%
20 or above	91	14.6%
**Place of Birth**		
Hong Kong	390	62.0%
Mainland China	204	32.4%
Other places	35	5.6%
**School/Faculty**		
Faculty of Business	1	0.2%
School of Design	4	0.6%
Faculty of Humanities	59	9.4%
Faculty of Construction and Environment	57	9.1%
Faculty of Engineering	228	36.2%
Faculty of Hotel and Tourism Management	47	7.5%
Faculty of Health and Social Sciences	106	16.9%
Faculty of Applied Science and Textiles	127	20.2%

**Table 2 ijerph-19-09809-t002:** Results of one-way repeated measures MANOVA (Students’ change between pre- and post-test) (*n* = 630).

Variables	Pre-Test		Post-Test			
	Mean (SD)	a	Mean (SD)	a	F Value	η^2^_p_
**Primary factors of Positive Youth Development**					18.17 **	0.261
Resilience	4.71 (0.87)	0.86	5.00 (0.87)	0.92	70.51 ***	0.101
Social Competence	4.69 (0.81)	0.90	5.00 (0.84)	0.94	75.63 ***	0.107
Emotional Competence	4.57 (0.79)	0.78	4.94 (0.84)	0.90	111.34 ***	0.150
Cognitive Competence	4.72 (0.72)	0.86	5.02 (0.81)	0.93	86.59 ***	0.121
Behavioural Competence	4.69 (0.77)	0.76	4.99 (0.83)	0.89	77.40 ***	0.110
Moral Competence	4.86 (0.74)	0.78	5.06 (0.80)	0.90	35.23 ***	0.053
Self-determination	4.57 (0.83)	0.83	4.93 (0.87)	0.89	92.76 ***	0.129
Self-efficacy	4.64 (0.84)	0.72	4.92 (0.92)	0.87	49.71 ***	0.073
Clear and Positive Identity	4.32 (0.91)	0.88	4.78 (0.92)	0.92	156.13 ***	0.199
Beliefs in the Future	4.79 (0.82)	0.87	5.01 (0.87)	0.91	42.68 ***	0.064
Prosocial Norms	4.92 (0.79)	0.76	5.10 (0.80)	0.83	25.30 ***	0.039
Spirituality	4.63 (0.92)	0.78	4.86 (0.95)	0.85	39.09 ***	0.059
**Higher-order Positive Youth Development factors**					32.28 ***	0.171
Cognitive-behavioural Competencies	4.67 (0.69)	0.92	4.99 (0.80)	0.96	102.14 ***	0.140
General PYD Qualities	4.68 (0.69)	0.94	4.97 (0.79)	0.97	87.06 ***	0.122
Positive Identity	4.52 (0.83)	0.92	4.88 (0.87)	0.95	114.58 ***	0.154
Total PYD	4.66 (0.68)	0.97	4.96 (0.78)	0.99	102.18 ***	0.140
**Wellbeing**					89.99 ***	0.223
Thriving	4.30 (0.67)	0.60	4.71 (0.82)	0.78	142.69 ***	0.185
Life Satisfaction	3.96 (0.98)	0.88	4.50 (1.08)	0.93	162.52 ***	0.205
**Desirable Graduate Attributes**					27.42 ***	0.150
Problem Solving	4.61 (0.75)	0.83	4.96 (0.85)	0.92	105.95 ***	0.145
Lifelong Learning	4.76 (0.80)	0.69	5.01 (0.83)	0.82	50.12 ***	0.074
Ethical Leadership	4.75 (0.63)	0.93	4.97 (0.74)	0.97	57.77 ***	0.084
Critical Thinking	4.76 (0.73)	0.88	5.03 (0.82)	0.94	66.61 ***	0.096

Note: ** *p* < 0.01; *** *p* < 0.001.

**Table 3 ijerph-19-09809-t003:** Descriptive statistics and frequencies of positive responses in subjective outcome evaluation.

Items	Participants with Positive Responses (Options 4–5)	Mean (SD)	*n*
**Student Perceptions of Course Design and Quality**	***n* (%)**		
1. The objectives of the curriculum are very clear.	553 (88.5%)	4.24 (0.74)	625
2. The design of the curriculum is very good.	509 (81.7%)	4.15 (0.83)	623
3. The activities were carefully arranged.	546 (86.9%)	4.22 (0.77)	628
4. The classroom atmosphere was very pleasant.	537 (85.6%)	4.24 (0.79)	627
5. There was much peer interaction amongst the students during classes.	523 (83.3%)	4.19 (0.82)	628
6. I participated actively during lessons (including discussions, sharing, games, etc.)	508 (81.0%)	4.12 (0.84)	627
7. I was encouraged to do my best.	535 (85.6%)	4.20 (0.77)	625
8. The learning experience I encountered enhance my interest towards the subject area.	504 (80.0%)	4.09 (0.91)	630
9. Overall speaking, I have very positive evaluation of the program.	528 (83.9%)	4.18 (0.84)	629
10. On the whole, I like this curriculum very much.	487 (77.8%)	4.04 (0.92)	626
**Student Perceptions of Lecturer Qualities**			
1. The lecturer has a good mastery of the curriculum.	578 (91.7%)	4.38 (0.72)	630
2. The lecturer was well prepared for the lessons.	590 (93.9%)	4.43 (0.68)	628
3. The teaching skills of the lecturer was good.	576 (92.2%)	4.38 (0.70)	625
4. The lecturer showed good professional attitudes.	593 (94.4%)	4.43 (0.67)	628
5. The lecturer was very involved.	594 (94.6%)	4.44 (0.67)	628
6. The lecturer encouraged students to participate in the activities.	588 (93.8%)	4.43 (0.67)	627
7. The lecturer cared for the students.	581 (93.0%)	4.42 (0.69)	625
8. The lecturer was ready to offer help to students when needed.	595 (94.9%)	4.44 (0.66)	627
9. The lecturer had much interaction with the students.	583 (93.0%)	4.38 (0.68)	627
10. Overall speaking, I have very positive evaluation of the lecturer.	587 (93.8%)	4.41 (0.68)	626
**Perceptions of the Course Benefits**			
1. The subject has strengthened my resilience in adverse conditions.	483 (77.0%)	4.05 (0.93)	627
2. The subject has enhanced my social competence.	498 (79.3%)	4.08 (0.90)	628
3. The subject has improved my ability in expressing and handling my emotions.	484 (77.1%)	4.05 (0.94)	628
4. The subject has enhanced my analytical ability.	486 (77.3%)	4.03 (0.93)	629
5. The subject has enhanced my critical thinking.	481 (76.5%)	4.04 (0.95)	629
6. The subject has strengthened my ability to distinguish between the good and the bad.	481 (76.7%)	4.03 (0.95)	627
7. The subject has increased my competence in making sensible and wise choices.	487 (77.7%)	4.06 (0.95)	627
8. This subject has helped me to have life satisfactions.	499 (79.6%)	4.13 (0.92)	627
9. This subject has strengthened my self-confidence.	459 (73.1%)	3.96 (1.01)	628
10. This subject has increased my self-awareness.	495 (78.7%)	4.07 (0.92)	629
11. This subject has helped me to face the future with a positive attitude.	483 (76.9%)	4.03 (0.98)	628
12. This subject has helped me to cultivate compassion and care about others.	488 (77.6%)	4.03 (0.97)	629
13. This subject has strengthened my motivation to learn something new every day.	477 (76.2%)	4.02 (0.97)	626
14. This subject has helped me to make ethical decision.	498 (79.0%)	4.09 (0.91)	630
15. This subject has enhanced my desire for lifelong learning to improve leadership competence.	490 (77.9%)	4.04 (0.97)	629
16. This subject has increased my ability to become an ethical leader.	492 (78.2%)	4.07 (0.96)	629
17. The theories, research and concepts covered in the course have enabled me to understand the characteristics of successful leaders.	509 (80.8%)	4.15 (0.90)	630
18. The theories, research and concepts covered in the course have helped me synthesize the characteristics of successful leaders.	507 (80.5%)	4.13 (0.89)	630
19. It has enabled me to understand the importance of interpersonal relationship in successful leadership.	503 (80.6%)	4.14 (0.88)	624
20. It has promoted my sense of responsibility in serving the society.	485 (77.4%)	4.06 (0.95)	627
21. It has enriched my overall development.	505 (80.4%)	4.11 (0.93)	628
**Willingness to recommend friends to take the course**	**Participants with Positive Responses (Options 3–4)**		
Will you suggest your friends to take this course?	520 (83.1%)	3.08 (0.77)	626
**Student Overall Satisfaction**	**Participants with Positive Responses (Options 4–6)**		
On the whole, are you satisfied with this course?	585 (92.9%)	4.70 (1.08)	630

For “Student Perceptions of Course Design and Quality” and “Students Perceptions of Lecturer Qualities”, all items are rated on a 5-point Likert-type scale with 1 = strongly disagree, 2 = disagree, 3 = neutral, 4 = agree, 5 = strongly agree. Only the positive responses (Options 4–5) are shown. For “Student Perceptions of the Course Benefits”, all items are rated on a 5-point Likert-type scale with 1 = unhelpful, 2 = not very helpful, 3 = slightly helpful, 4 = helpful, 5 = very helpful. Only the positive responses (Options 4–5) are shown. The item “Will you suggest your friends to take this course?” is rated on a 4-point Likert-type scale with 1 = definitely will not suggest, 2 = will not suggest, 3 = will suggest, 4 = definitely will suggest. Only the positive responses (Options 3–4) are shown. The item “On the whole, are you satisfied with this course?” is rated on a 6-point Likert-type scale with 1 = very dissatisfied, 2 = moderately dissatisfied, 3 = dissatisfied, 4 = satisfied, 5 = moderately satisfied, 6 = very satisfied. Only the positive responses (Options 4–6) are shown.

**Table 4 ijerph-19-09809-t004:** Impact of perceived course design, lecturer quality and course benefits on overall satisfaction.

Predictors	Overall Satisfaction		
**Step 1**	**β**	** *t* **	**Cohen’s *f*^2^**
Gender	0.09	2.22 *	0.008
Age	−0.07	−1.83	0.003
Place of Birth	0.27	6.87 ***	0.076
Faculty	−0.001	−0.04	0.000
*R^2^*	0.084		
*F*	14.10 ***		
**Step 2**	**β**	** *t* **	**Cohen’s *f*^2^**
Perceived Course Design	0.45	7.77 ***	0.098
Perceived Lecturer Quality	0.13	3.43 **	0.020
Perceived Course Benefits	0.27	5.45 ***	0.049
*R^2^*	0.693		
*R^2^* Change	0.610		
*F*	198.30 ***		

Note: * *p* < 0.05; ** *p* < 0.01; *** *p* < 0.001.

**Table 5 ijerph-19-09809-t005:** Predicting effect of perceived course design, lecturer, and course benefits on post-test PYD after controlling pre-test PYD and demographic factors.

Predictors	Post-Test PYD		
**Step 1**	**β**	** *t* **	**Cohen’s *f^2^***
Gender	0.04	1.15	0.003
Age	−0.001	−0.02	0.000
Place of Birth	0.13	3.46 **	0.019
Faculty	0.02	0.55	0.000
Pre-test PYD	0.47	12.92 ***	0.271
*R^2^*	0.267		
*F*	44.85 ***		
**Step 2**	**β**	** *t* **	**Cohen’s *f^2^***
Perceived Course Design	0.21	2.93 **	0.013
Perceived Lecturer Quality	0.12	2.64 **	0.011
Perceived Course Benefits	0.26	4.24 ***	0.028
*R^2^*	0.528		
*R^2^ Change*	0.262		
*F*	85.88 ***		

Note: ** *p* < 0.01; *** *p* < 0.001.

## Data Availability

The data presented in this study are available on request from the corresponding author.

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
