# Peer review of "The Effectiveness of a Leadership Subject Using a Hybrid Teaching Mode during the Pandemic: Objective Outcome and Subjective Outcome Evaluation"

_ijerph, 2022, doi:10.3390/ijerph19169809_

Round 1

Reviewer 1 Report

The paper is well organized and well written. The research of the problem is well designed as well as the method. Finally the results and conclusions present the main findings and limitations of the work.

Author Response

Point 1: The paper is well organized and well written. The research of the problem is well designed as well as the method. Finally, the results and conclusions present the main findings and limitations of the work.

Response 1: We thank the reviewer for his/her encouraging comments.

Reviewer 2 Report

Authors claim that this study produced positive results, objective and subjective including both objective and subjective outcome evaluation.

It can be true but it is not clearly grasped from the text, as it is. It presents several challenges to be overcome in order to really allow readers to fully understand what they are proposing in the paper.

First, huge tables full of numbers are not readable at all. 

Table 2 is encrypted, nearly impossible to read and understand it without following more than carefully the text.

Table 3 appears after Table 5. Is that OK?

Table 3 is HUGE and highly complex. I wonder its real value to readers. Any chance to simplify its content? Otherwise, I’m afraid data is not useful in this form. Maybe authors shall use graphical forms using more tangible axis instead of the long list of parameters and values.

Second and most important, authors stated in the conclusions that “the study adopted 479 one group pre-test and post-test design in objective outcome evaluation, 

which may have bias arising from the absence of a control group since the participants’ improvement may be due to other factors …”. A control group seems fundamental in order to support with no bias this research, which means a considerable flaw at this stage of the research. 

Paragraph in section 3.1 is unreadable. To confuse, too many numbers. The same problem is found in section 3.2. 

Authors shall find a way to explain their results in a more clean and clear way.

Several concepts just popup without a proper explanation (e.g. the Cronbach’s alpha, “one-way re-peated measures MANOVA was conducted”, “SPSS Version 25”) which brings some difficulty to readers.

English shall be improved.

Minor problems like the use of “e.g.” in the references (“[e.g., 55,56]”)

Reviewer 3 Report

Der authors,

I found the paper very interesting, sound and well designed. The results are valuable and of a broad interest. I have some minor comments:

1) The title doesn't fully reflect the research findings. The term 'Change in students ...' is not clear. Also, 'taking a leadership subject under covid-19', presents a strange structure. It should be fully rewritten.

2) I do not think that the background with socioeconomic status drives mental health directly. Lines 33-34 shoud be rewritten.

3) The results of Table 5 should be presented with more details. The description is too short.

4) Line 417: should be programs.
